# Ketamine for critically ill patients with severe acute brain injury: Protocol for a systematic review with meta-analysis and Trial Sequential Analysis of randomised clinical trials

Frederik Andreas Madsen[1]⊛, Trine Hjorslev Andreasen[2,3]⊛*, Jane Lindschou[4], Christian Gluud[4,5], Kirsten Møller[1,3]

**1** Department of Neuroanaesthesiology, Neuroscience Centre, Copenhagen University Hospital—Rigshospitalet, Copenhagen, Denmark, **2** Department of Neurosurgery, Neuroscience Centre, Copenhagen University Hospital—Rigshospitalet, Copenhagen, Denmark, **3** Department of Clinical Medicine, Faculty of Health and Medical Sciences, University of Copenhagen, Copenhagen, Denmark, **4** Copenhagen Trial Unit, Centre for Clinical Intervention Research, The Capital Region, Copenhagen University Hospital—Rigshospitalet, Copenhagen, Denmark, **5** Department of Regional Health Research, Faculty of Health Sciences, University of Southern Denmark, Odense, Denmark

⊛ These authors contributed equally to this work.
* trine.hjorslev.andreasen@regionh.dk

## ● OPEN ACCESS

**Funding:** The authors received financial support from the Department of Neurosurgery Research Pool (Neurokirurgisk forskningspulje) at

## Abstract

### Introduction

Intensive care for patients with severe acute brain injury aims both to treat the immediate consequences of the injury and to prevent and treat secondary brain injury to ensure a good functional outcome. Sedation may be used to facilitate mechanical ventilation, for treating agitation, and for controlling intracranial pressure. Ketamine is an N-methyl-D-aspartate receptor antagonist with sedative, analgesic, and potentially neuroprotective properties. We describe a protocol for a systematic review of randomised clinical trials assessing the beneficial and harmful effects of ketamine for patients with severe acute brain injury.

### Methods and analysis

We will systematically search international databases for randomised clinical trials, including CENTRAL, MEDLINE, Embase, and trial registries. Two authors will independently review and select trials for inclusion, and extract data. We will compare ketamine by any regimen versus placebo, no intervention, or other sedatives or analgesics for patients with severe acute brain injury. The primary outcomes will be functional outcome at maximal follow up, quality of life, and serious adverse events. We will also assess secondary and exploratory outcomes. The extracted data will be analysed using Review Manager and Trials Sequential Analysis. Evidence certainty will be graded using GRADE.

### Ethics and dissemination

The results of the systematic review will be disseminated through peer-reviewed publication. With the review, we hope to inform future randomised clinical trials and improve clinical practice.

Rigshospitalet, Master Carpenter Sophus Jacobsen and wife Astrid Jacobsen's Foundation (Snedkermester Sophus Jacobsen og hustru Astrid Jacobsen's Fond), Rigshospitalet's 3-year PhD scholarship, and the Danish Victims Foundation (20-610-00103) in the form of salary for author TA. Funding was also received from: The A.P. Møller Foundation (20-L-0041), Knud and Edith Eriksen's Memorial Fund (Knud og Edith Eriksens Mindefond, 62786-2021), and The Novo Nordisk Foundation (NNF20OC0065750). The funders had no role in study design, data collection and analysis, decision to publish, or preparation of the manuscript.

**Competing interests:** The authors have declared that no competing interests exist.

## PROSPERO no

CRD42021210447.

## Introduction

Intensive care for patients with severe acute brain injury often includes administration of sedative and analgesic medications. The use of ketamine, which has both sedative and analgesic properties, reportedly lowers the incidence of cortical spreading depolarisation, a pathophysiological phemomenon that may cause secondary brain injury [1–6]. Existing guidelines do not specify sedative and analgesic agents of choice in patients with severe acute brain injury. Propofol and midazolam are currently the most frequently used drugs in this setting [6–10]. Ketamine is not commonly used [8]. Here we describe a protocol for a systematic review, which explores potential benefits and harms of ketamine for treating critically ill patients with severe acute brain injury.

### Description of the patient population

Encompassing traumatic brain injury, subarachnoid haemorrhage, intracerebral haemorrhage, acute ischaemic stroke, hypoxic brain injury after cardiac arrest [11–20], as well as complications from neurosurgery [21], severe acute brain injury is a common cause of critical illness requiring admission to intensive or neurointensive care units [11, 22–24].

Severe acute brain injury is associated with prolonged intensive care unit and hospital stay, permanent disability, low quality of life, and death [17]. These outcomes depend on both the primary brain injury, which is immediate and irreversible, and the ensuing secondary brain injury, which is potentially reversible [18, 25–27] and induced by triggers such as hypoxia, hypotension, and inflammation [12, 13, 18, 28, 29].

Aneurysmal subarachnoid haemorrhage, which occurs in approximately 9 persons per 100,000 population-years at risk globally, has a fatality of around 25% with 20% of all patients remaining functionally dependent on help for activities of daily living [20, 30]. Early-phase complications such as rebleeding and delayed cerebral ischaemia are important predictors of long-term outcome [12, 31]. Early aneurysm closure may prevent rebleeding, but the mechanisms leading to delayed cerebral ischaemia are poorly understood and there is currently no known effective treatment [27]. Traumatic brain injury warranting admission to hospital (though not necessarily to the intensive care unit) occurs in 287 per 100,000 population-years at risk in Europe and is the cause of death in 37% of all trauma-related deaths [32]. Traumatic brain injury leads to a wide range of long-term disabilities that affects functional outcomes in survivors. Hypotension or hypoxia early after the injury are notorious risk factors for a poor outcome [33, 34]. Spontaneous intracerebral haemorrhage occurs in 25 per 100,000 population-years at risk globally and has a one-year mortality of 60%. Only 12% to 39% of afflicted patients will achieve functional independence [35]. Haematoma growth during the early phase is widely accepted as the most important risk factor for death and a poor outcome [35, 36].

Treatment of severe acute brain injury focuses on limiting the consequences of the primary injury and preventing secondary injury [37]. Administration of nimodipine, for example, may reduce the risk of delayed cerebral ischaemia and a poor functional outcome in patients with aneurysmal subarachnoid haemorrhage [38]. Moreover, multiple parameters such as intracranial pressure (ICP), cerebral metabolism (through microdialysis), cerebral electrical activity (measured using electroencephalography or electrocorticography), and brain-tissue

oxygenation (PbtO$_2$) [11, 28, 39] may be monitored to support the clinical neurological examination for early identification of known complications to brain injury [28].

## Description of the intervention

Ketamine is a phenylcyclidine derivate with two optical enantiomers [40]. It is commercially available either as a racemic mixture, or as the (S)-enantiomer alone, which features approximately twice the potency of the racemic mixture [41], with potentially fewer psychomimetic adverse effects [42]. The main mechanism of action of ketamine has been attributed to NMDA-receptor (N-methyl-D-aspartate-receptor) antagonism [43]. The NMDA receptor is an ionotropic glutamate receptor. Activation of the receptor requires two distinct stimuli with temporal proximity–glutamate binding and membrane depolarisation [44]. Even though ketamine has a direct negative inotropic and chronotropic effect on the heart, ketamine has an overall sympathomimetic effect resulting in increased heart rate, cardiac output, and mean arterial blood pressure [45]. It is often stated that the cardiovascular stability of ketamine makes it a favourable choice in haemodynamically compromised patients, although there have been voices of dissent [46].

Featuring sedative, amnestic, and analgesic properties, ketamine has the potential to be the sole agent for induction and maintenance of anaesthesia [40]. Furthermore, protective airway reflexes, such as pharyngeal and laryngeal reflexes, are generally preserved and only negligible respiratory depression is reported [40]. Due to its sympathomimetic and bronchodilating effects, ketamine has found favour internationally as a sedative hypnotic agent in patients with haemodynamic instability and in those with reactive airway disease. In the prehospital setting, ketamine is used for rapid sequence induction prior to intubation as well as for pain management, in particular in patients with hypovolaemic shock as well as in war and combat zones [47]. In developing countries, ketamine is used as a simple and inexpensive alternative to a combinatorial drug approach that typically involves a concoction of more costly sedatives and analgesics [48]. Other notable clinical uses of ketamine include sedation of children and burn victims [47].

A number of national guidelines on intensive care recommend only light sedation during mechanical ventilation [7–9], though deep sedation is indicated for a series of specific clinical conditions, including increased intracranial pressure [6, 7, 9]. The choice of sedative is guided by indications and sedation goals, though propofol and midazolam are the standard sedatives used, and none of the guidelines suggests the use of ketamine specifically [7–10].

## How the intervention might work

Subarachnoid haemorrhage, traumatic brain injury, acute ischaemic stroke with massive hemispheric infarction, and spontaneous intracerebral haemorrhage with postoperative perihaematomal oedema progression have all been associated with the presence of cortical spreading depolarisation [49–52]. These are slowly migrating depolarisation waves in the cerebral cortex, which may be monitored in patients using electrocorticography [53] or, potentially, electroencephalography [54, 55]. Repolarisation after such depolarisations dramatically increases brain metabolism [56]. The haemodynamic response to cortical spreading depolarisation often involves prolonged hypoperfusion [57], which may result in ischaemia and hypoxia in vulnerable brain tissue [57]. Cortical spreading depolarisations are associated with an increased risk of an unfavourable outcome after severe acute brain injury [58] and may thus represent a potential target for intervention. Ketamine appears to inhibit cortical spreading depolarisations in both animal and human studies [1–3, 5], has been suggested to attenuate

excitotoxicity, and has anti-inflammatory and anti-apoptotic effects [59]. Ketamine administration could therefore be advantageous in critically ill patients with severe acute brain injury.

## Why it is important to do this systematic review

Despite its multiple potential benefits, concerns of increased ICP has limited the use of ketamine in patients with severe acute brain injury [8, 60]. Yet recent analyses suggest that ICP-related reservations about ketamine may be unfounded [61, 62]. Today, the use of ketamine in the ICU is an emerging area of interest [6]. Thus, in vitro studies [2, 63], patient series [1, 3, 4, 53], and a cross-over study [5] indicate that ketamine compared to other types of sedatives reduces the incidence of cortical spreading depolarisation in patients with acute brain injury (primarily traumatic brain injury and subarachnoid haemorrhage). A previous systematic review of sedatives and analgesics in critically ill patients with traumatic brain injury found no randomised clinical trials that addressed the effect of ketamine compared with other sedatives or analgesics on clinical outcomes [64]. Two randomised trials [65, 66] investigated the effect of ketamine versus the drug sufentanil on intracranial pressure and cerebral perfusion pressure, reporting no differences in these variables.

One systematic review examined the use of ketamine for patients with traumatic brain injury, focusing mainly on ICP, and concluded that the overall evidence of using ketamine for this patient population is low [67].

In conclusion, ketamine may potentially improve outcome after acute brain injury, but the existing literature on its benefits and harms has not been sufficiently examined previously.

## Objectives

The objective of this systematic review is to assess benefits and harms of ketamine for critically ill patients with severe acute brain injury.

## Methods and analysis

In this systematic review, we will use standard methodological procedures following the recommendations by the Cochrane Handbook for Systematic Reviews of Interventions [68]. We will use the Grading of Recommendations Assessment, Development and Evaluation (GRADE) approach for grading evidence of the outcomes of interest [69], and the eight-step approach proposed by Jakobsen and colleagues to assess the intervention effect [70]. This protocol will adhere to the Preferred Reporting Items for Systematic Review and Meta-Analysis (PRISMA-P) checklist [71]. Likewise, the upcoming systematic review will be reported according to the PRISMA 2020 guidelines [72]. Inclusion and exclusion of trials will be visualised in a 2020 PRISMA flow diagram [72].

The protocol is registered in the PROSPERO database with registration no. CRD42021210447.

## Ethics and dissemination

There are no ethical or safety considerations related to this systematic review. The results from the upcoming systematic review will be disseminated through publication in a peer-reviewed journal.

## Criteria for considering studies for inclusion

**Types of studies.** We will search for and include randomised clinical trials and cross-over randomised clinical trials as well as cluster-randomised trials for the assessments of benefits

and harms, irrespective of reported outcomes, publication date, publication language, publication type, and publication status.

For the assessment of harms, we will also include quasi-randomised studies and observational studies, that are identified during our searches for trials. Such studies will be reported narratively and separately.

**Participants.**  Studies of patients of all ages with severe acute brain injury treated in the ICU are included. Severe acute brain injury is defined as aneurysmal subarachnoid haemorrhage, traumatic brain injury, acute ischaemic stroke, intracerebral haemorrhage, or hypoxic brain injury.

**Interventions.**  We will include trials in which ketamine administration is compared with either placebo, 'no intervention', or drugs that are used as standard of care in the ICU (e.g. propofol, benzodiazepines, opioids, alpha-2-agonists, and antipsychotics). No limitations to dose, formulation, or treatment duration will be made. Racemic mixtures of ketamine as well as S-ketamine will be included. Trials investigating ketamine administration for procedural sedation only will be excluded.

**Types of outcomes.**  The ideal assessment time depends on the different outcomes. We will specify the time points below.

*Primary outcomes.*

- Proportion of participants with unfavourable functional outcome at maximal follow up. We will evaluate functional outcome by the dichotomised Glasgow Outcome Scale (GOS), Glasgow Outcome Scale—Extended (GOSE), or modified Rankin Scale (mRS) and through them categorise a patient to unfavourable or favourable functional outcome. Unfavourable outcome is defined by the trial authors or by a GOS score of 1–3, GOSE score of 1–5, or mRS score of 3–6.

- Quality of life at maximal follow up as measured using any validated scale.

- Proportion of participants with one or more serious adverse events at maximal follow up. These are defined according to the International Conference on Harmonization (ICH) Guidelines (ICH-GCP 1997) [73], that is, any event that leads to death; is life-threatening; requires in-patient hospitalisation or prolongs an existing hospital stay; or results in persistent or significant disability, congenital birth, or anomaly; and any important medical event that may have jeopardised the patient or required intervention to prevent it. All other adverse events are classified as non-serious.

*Secondary outcomes.*

- All-cause mortality at maximal follow-up.

- The proportion of participants with neurological complications (rebleeding, seizures, worsening of neurological deficits, etc.) at maximal follow up.

- Reported agitation during the ICU stay, as measured by validated sedation scales such as the Richmond Agitation Sedation Scale (RASS) or the Ramsay Sedation Scale, during administration of experimental and control interventions.

- Proportion of participants with one or more adverse event (dichotomous data) during the hospital stay that is not considered serious.

*Exploratory outcomes.*

- Functional clinical outcome defined by authors or evaluated with a valid neurological outcome scale, e.g. GOS, GOSE, or mRS at maximal follow-up.

- Reported pain during the ICU stay during administration of experimental and control interventions (number of patients and frequency).

- ICU length of stay (in days).

- Duration of mechanical ventilation (in days).

- Proportion of participants with individual serious adverse events at maximal follow up.

- Proportion of participants with individual adverse events not considered serious during the hospital stay.

- Multimodal neuromonitoring parameters during the ICU stay, such as (but not restricted to) ICP, $PbtO_2$, and electrocorticographic spreading depolarisations, as defined by the trial authors.

## Search methods for identification of studies

**Databases.** The following databases will be searched for published studies: MEDLINE, Embase, CENTRAL, CINAHL, LILACS, and Web of Science. For unpublished studies, we will search: ClinicalTrials.gov, WHO International Clinical Trials Registry Platform (ICTRP), and EU Clinical Trials Register. The full search strategy can be found in S1 File.

**Other sources.** Reference lists of relevant papers will be screened for further studies or randomised clinical trials not found in the database searches.

## Data collection and analysis

**Selection of studies.** Two authors (FM and THA) will independently select studies, perform data extraction, and assess risks of bias. Any disagreement will be resolved by consensus, with a third author (KM) as the final arbiter. Titles and abstracts of all reports identified by the searches are screened. Potentially relevant reports are obtained in full text and are assessed for inclusion.

**Data extraction and management.** Two authors (FM and THA) will independently extract predefined data of included reports using a data collection form designed by the review team. Corresponding authors will be contacted for further information if there are issues related to data reporting or missing data.

**Assessment of risk of bias in included randomised clinical trials.** Two authors (FM and THA) will independently assess the included trials for risks of bias using the revised Cochrane risk of bias tool for randomised trials (RoB 2) [74]. The following five domains are evaluated: 1) bias arising during the randomisation process, 2) bias due to deviations from intended interventions, 3) bias due to missing outcome data, 4) bias in measurement of the outcome, and 5) bias in selection of the reported result.

**Measures of treatment effect.** The primary outcome of the present protocol is functional clinical outcome, and this is usually assessed using a multi-step (ordinal) scale with perfect neurological outcome at one end and death at the other end. For practical purposes, in clinical trials this multi-step assessment is frequently converted into a dichotomous outcome ('favourable outcome' and 'unfavourable outcome', or similar terminology). However, the original ordinal scales contain more information and have been reported to correspond better to long-term functional outcomes, at least for ischaemic stroke [75].

We will use an intention-to-treat analysis to make comparisons, where possible. For proportions (dichotomous outcomes), we will use relative risks (RRs) with 95% confidence intervals (CI) and Trial Sequential Analysis-adjusted CI. To analyse ordinal scale scores, we will use

several analytic methods to test the robustness of our data. First, we will analyse the scales as dichotomised scales (functional outcomes: GOS 1–3 compared to 4–5, GOSE 1–5 compared to 6–8, mRS 0–2 compared to 3–6). Second, we will analyse the scales as continuous data with mean differences with 95% CIs and Trial Sequential Analysis-adjusted CI. Third, we will consider using the proportional odds model, if possible [68]. We will consider the results of all three analyses in the interpretation of effects and directions of effects. Finally, we will calculate the number needed to treat (NNT) and number needed to harm (NNH) with 95% CI where appropriate.

**Unit of analysis issues.**   In case of multi-group trials comparing different drugs, including ketamine, as interventions, we will not combine the other drugs into one intervention group and compare this group to the ketamine group. The analyses will be conducted independently so that conclusions may be drawn.

Cluster-randomised trials, if detected, are included for analysis provided that they can be reduced to their effective sample sizes (with regard to the intracluster correlation).

For cross-over trials, we will include only outcomes from the end of the first period of the trial where cross-over trials resemble usual randomised clinical trials, to avoid bias arising from carry-over effects.

We will analyse additional domains of bias in cluster-randomised trials and cross-over trials in accordance with the recommendations presented in the Cochrane Handbook for Systematic Reviews of Interventions [68].

**Missing data.**   We will contact investigators to retrieve important missing data. This applies to missing data for our primary, secondary, and exploratory outcomes, as well as methodological information for the risk of bias assessment. Additionally, we will seek information on the relevant subgroup(s) in case of trials with a mixed population. In case of no reply, we will make a second and third attempt to get in contact with the authors in question. If the missing data is irretrievable, we will take the percentage of missing data into account when interpreting the results (see below). Standard deviations (SD), if not reported by authors, will be calculated using data from the trial if possible.

**Assessment of heterogeneity.**   We will assess heterogeneity by visual inspection of the forest plots, by performing chi-squared test with a $P < 0.10$ as statistical level of significance, and by measuring $I^2$. According to the Cochrane Handbook for Systematic Reviews of Interventions (Version 6, 2019), heterogeneity is described as considerable if $I^2$ is between 75% to 100%, substantial between 50% to 90%, moderate between 30% to 60%, and nonexistent or low between 0% to 40%.

**Assessment of reporting bias.**   We will visually examine funnel plots for signs of asymmetry if ten or more trials are included in an analysis [70].

## Data synthesis

**Meta-analysis.**   We will conduct meta-analysis based on recommendations in the Cochrane Handbook for Systematic Reviews of Interventions [68] if effect measures from two or more trials are comparable and the clinical heterogeneity is not too extensive. If the results are not suitable for quantitative analysis, they will be reported narratively.

We will report results from both fixed-effect and random-effects meta-analyses. If they show differing results, we will choose the one with the most conservative result (highest P-value and widest confidence interval) as the main result. For these analyses, we will make use of *Review Manager*, a statistical software program offered by The Cochrane Collaboration.

**Assessment of significance.**   We will follow the eight-step approach proposed by Jakobsen et al. [70] for evaluating statistical and clinical significance of the meta-analysis. This includes

fixed-effect meta-analysis, random-effects meta-analysis, subgroup analysis, sensitivity analysis, adjustments of the thresholds for significance, calculations of realistic diversity-adjusted required information size, Trial Sequential Analysis, calculations of Bayes factor, assessment of the impact of bias, assessment of publication bias, and assessment of clinical significance [70].

To control problems with multiplicity, we will adjust the significance thresholds by a modified Bonferroni adjustment, dividing the threshold by the value halfway between 1 (no adjustment) and the number of primary or secondary outcomes, respectively [70]. Consequently, the threshold of statistical significance will be 0.025 for our three primary outcomes and 0.02 for our four secondary outcomes. For exploratory outcomes, we will use P < 0.05, as we only consider these analyses as hypothesis generating.

**Trial Sequential Analysis.** If meta-analysis is appropriate, we will conduct Trial Sequential Analysis (TSA) to control the risk of random errors due to sparse data and repetitive testing [76]. Both primary and secondary outcomes will be analysed. TSA of the meta-analysis will be performed by using the TSA software provided by The Copenhagen Trial Unit [77] and will yield the required information size to obtain definite evidence on the effect of an intervention, an adjusted confidence interval, and an adjusted level of statistical significance [76].

For dichotomous outcomes, we will estimate the required information size based on the observed proportion of patients with an outcome in the control group of the meta-analysis (the cumulative proportion of patients with an event in the control groups relative to all patients in the control groups), a relative risk reduction or increase of 20%, an alpha of 2.5% for all primary outcomes, a beta of 10%, and the observed diversity as suggested by the trials in the meta-analysis. For continuous outcomes, we will use the observed SD, a mean difference of the observed SD/2, an alpha of 2.5% for all primary outcomes, a beta of 10%, and the observed diversity as suggested by the trials in the meta-analysis. For the secondary outcomes we will use an alpha of 2% plus similar parameters as to the primary outcomes.

**Subgroup analysis and investigation of heterogeneity.** In order to assess whether a given effect is present in only a subgroup of trials, patients, or dosage regimens, or whether the effect is influenced by the type of comparator, we will conduct the following comparisons between estimated intervention effects, if possible:

- Trials at overall 'low' compared to trials at 'high' risk of bias

- Different patient diagnoses (aneurysmal subarachnoid haemorrhage, traumatic brain injury, acute ischaemic stroke, intracerebral haemorrhage, and hypoxic brain injury)

- Different age groups (infants, children, adolescents, and adults)

- Doses of ketamine at or higher compared to lower than the median dose

- Treatment duration at or longer compared to shorter than the median duration

- Different comparators ('no intervention', placebo, other sedatives, or analgesics)

**Sensitivity analysis.** In line with the eight step approach [70], we will conduct sensitivity analyses to assess the potential impact of missing outcome data. Thus, dichotomous outcomes will undergo two sensitivity analyses on both the primary and secondary outcomes, i.e. 'best-worst-case' and 'worst-best-case' scenarios [70]. Continuous outcomes in the best group, e.g. quality of life, will be imputed as the group mean plus two standard deviations (SDs) of the group mean. Worst groups are imputed as the group mean minus two SDs of the group mean. We will present results of all scenarios in our review. Other post-hoc sensitivity analyses might be warranted if unexpected clinical or statistical heterogeneity is identified during the analysis of the review results.

**Summary of findings.** We will assess the quality of primary and secondary outcomes and the certainty of evidence with GRADE [69]. We will construct summary of findings tables in the GRADEpro GDT software [78] including the following outcomes:

1. The proportion of participants with unfavourable functional clinical outcome at maximal follow

2. Quality of life at maximal follow

3. The proportion of participants with one or more serious adverse events

4. All-cause mortality at maximal follow-up.

5. The proportion of participants with neurological complications (rebleeding, seizures, worsening of neurological deficits, etc.)

6. Reported agitation during the ICU stay, as measured by validated sedation scales such as the Richmond Agitation Sedation Scale (RASS) or the Ramsay Sedation Scale, during administration of experimental and control interventions.

7. Proportion of participants with at least one adverse event (dichotomous data) during the hospital stay that is not considered serious

## Discussion

We aim to explore the benefits and harms of ketamine for critically ill patients with severe acute brain injury based on conducted randomised clinical trials. Because these patients have a high risk of poor outcomes such as death and permanent disability [17], there is a need to improve the current standard of care. According to recent studies, ketamine may prevent secondary brain injury due to cortical spreading depolarisation [1–3, 5], one of several potentially deleterious mechanisms [59].

Concerning strengths, this protocol is made in accordance with PRISMA-P [71] and the Cochrane Handbook for Systematic Reviews of Interventions [68]. To further strengthen our review, we will use Trial Sequential Analysis [76, 77], the GRADE methodology [69], and the eight-step assessment suggested by Jakobsen et al. [70].

Concerning weaknesses, first, we have decided only to search for randomised clinical trials, cross-over trials and cluster-randomised trials, and not for quasi-randomised studies and observational studies. Quasi-randomised studies and observational studies that are identified during our searches, e.g. as important references cited in manuscripts of randomised, cross-over and cluster-randomised trials, will however be included for a crude assessments of harms. This means that we may overlook late or rare adverse effects, which may necessitate further review of quasi-randomised studies and observational studies in the future [79]. Second, we intend to dichotomise ordinal outcomes, at the risk of losing statistical power. Any results on unfavourable functional clinical outcome at maximal follow up should be interpreted with caution for a number of reasons [80]: 1) the assessments of unfavourable functional clinical outcome is based on single scores and it is questionable whether single scores are indications of unfavourable functional clinical outcome to the intervention; 2) information is lost when continuous data are transformed to dichotomous data and the analysis results can be greatly influenced by the distribution of data and the choice of an arbitrary cut-point; 3) even though a larger proportion of participants cross the arbitrary cut-point in the experimental group compared with the control group, the effect measured might still be limited; 4) by only focusing on how many patients cross a certain line for poor outcome, investigators ignore how many

patients are improving at the same time. The clinical significance of our results on functional outcome may therefore be questioned. According to Kirsch and Moncrieff, "response rates based on continuous data do not add information, and they can create an illusion of clinical effectiveness" [81].

We hope that this systematic review may elucidate possible beneficial and harmful effects of ketamine in critically ill patients with severe acute brain injury. This may lead the way for future good-quality randomised clinical trials and evidence-based clinical practise.

## Supporting information

**S1 Checklist. PRISMA-P 2015 checklist.**
(DOCX)

**S1 File. Search strategy.**
(DOC)

## Acknowledgments

The authors acknowledge information specialist Sarah Louise Klingenberg for assistance with the search strategy.

## Author Contributions

**Conceptualization:** Frederik Andreas Madsen, Trine Hjorslev Andreasen, Kirsten Møller.

**Investigation:** Frederik Andreas Madsen, Trine Hjorslev Andreasen.

**Methodology:** Frederik Andreas Madsen, Trine Hjorslev Andreasen, Jane Lindschou.

**Supervision:** Jane Lindschou, Christian Gluud, Kirsten Møller.

**Writing – original draft:** Frederik Andreas Madsen, Trine Hjorslev Andreasen.

**Writing – review & editing:** Frederik Andreas Madsen, Trine Hjorslev Andreasen, Christian Gluud, Kirsten Møller.

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
