## [Decision Letter · Decision Letter 0]

31 Aug 2021

PONE-D-21-17647

Ketamine for critically ill patients with severe acute brain injury: protocol for a systematic review with meta-analysis and Trial Sequential Analysis of randomised clinical trials

PLOS ONE

Dear Dr. Andreasen,

Thank you for submitting your manuscript to PLOS ONE. After careful consideration, we feel that it has merit but does not fully meet PLOS ONE’s publication criteria as it currently stands. Therefore, we invite you to submit a revised version of the manuscript that addresses the points raised during the review process.

I apologize for the delay in providing a decision for your work. I am the second Associate Editor managing your submission; I was invited on July, 27th.

The work is timely and the topic is of great interest for multiple medical specialities. The methodology is sound and described in details.

We look forward to receiving your revised manuscript.

Kind regards,

Alessandro Putzu, M.D.

Academic Editor

PLOS ONE

Journal Requirements:

5. Thank you for submitting the above manuscript to PLOS ONE. During our internal evaluation of the manuscript, we found significant text overlap between your submission and the following previously published works, some of which you are an author.

https://systematicreviewsjournal.biomedcentral.com/articles/10.1186/s13643-019-0957-0

https://link.springer.com/chapter/10.1007%2F978-981-13-7272-8_1

Please revise the manuscript to rephrase the duplicated text, cite your sources, and provide details as to how the current manuscript advances on previous work. Please note that further consideration is dependent on the submission of a manuscript that addresses these concerns about the overlap in text with published work.

Additional Editor Comments:

Minor comments:

1- I suggest to include a PRISMA 2020 checklist and a PRISMA 2020 flow-chart in your systematic review.

2- Please report more details on missing outcome data. What does “important missing data” mean? You will contact corresponding authors only for primary outcome data? What about, for example, secondary outcomes or methodological information for risk of bias? Probably you should define a priori some important missing data to be asked to corresponding authors. How many time will you contact corresponding authors in case of no reply?

Reviewers' comments:

Reviewer's Responses to Questions

**Comments to the Author**

1. Does the manuscript provide a valid rationale for the proposed study, with clearly identified and justified research questions?

Reviewer #1: Yes

Reviewer #2: Yes

Reviewer #3: Yes

2. Is the protocol technically sound and planned in a manner that will lead to a meaningful outcome and allow testing the stated hypotheses?

Reviewer #1: Yes

Reviewer #2: No

Reviewer #3: Yes

3. Is the methodology feasible and described in sufficient detail to allow the work to be replicable?

Reviewer #1: Yes

Reviewer #2: Yes

Reviewer #3: Yes

4. Have the authors described where all data underlying the findings will be made available when the study is complete?

Reviewer #1: Yes

Reviewer #2: Yes

Reviewer #3: Yes

5. Is the manuscript presented in an intelligible fashion and written in standard English?

Reviewer #1: Yes

Reviewer #2: Yes

Reviewer #3: Yes

6. Review Comments to the Author

You may also provide optional suggestions and comments to authors that they might find helpful in planning their study.

Reviewer #1: This proposed study protocol has sound methodology. This study proposal is certainly a question worth asking, with important clinical implications that could change the way we manage this vulnerable subset of patients.

Some small changes to the language used may make it easier to follow. I've included some examples below:

Suggestions for the Introduction:

Have faith in your audience. Most would agree that severe acute brain injury generally has two initial pathways, admission to ICU or death. It doesn't add much to say they may be admitted to ICU. Instead your intro may be more effective if the first line read: "Intensive care for patients with severe acute brain injury often includes administration of sedative and analgesic medications."

The sentence beginning with "Existing guidelines" probably shouldn't read "do not recommend," as this may also be interpreted as "recommend against," which is not your intent. Instead you may consider: "Existing guidelines do not specify sedative and analgesic agents of choice in patients with severe acute brain injury. Propofol and midazolam are currently the most frequently used drugs in this setting."

Suggestions for "Description of the Patient Population"

Consider consolidating the first 2 sentences in this section: "Encompassing traumatic brain injury, subarachnoid haemorrhage, intracerebral haemorrhage, acute ischaemic stroke, hypoxic brain injury after cardiac arrest, as well as complications from neurosurgery, Severe acute brain injury is a common cause of critical illness requiring admission to intensive or neurointensive care units."

paragraph 2: substitute "prolonged" for "long durations of"

paragraph 3: You already described the range of specific diseases that severe acute brain injury encompasses. Consider removing the intro sentence beginning "the specific diseases." and delete the words "as examples." Instead begin the paragraph with "Aneurysmal subarachnoid hemorrhage, which occurs...."

after "long-term outcome", "Early aneurysm closure..." should be it's own new sentence (not a semicolon)

After "60%" consider ending the sentence and starting a new sentence. Consider "Only 12% to 39% of afflicted patients will achieve functional independence."

Paragraph 4 (Treatment):

2nd sentence reads awkwardly. Consider instead, "Administration of nimodipine, for example, may reduce the risk of delayed cerebral ischemia..."

Suggestions for "Description of the intervention"

Paragraph 1: I would spend a little more time expanding here. (some physicians may also be less familiar with S-ketamine so there is an opportunity to educate. As written, it starts out a bit unclear when describing the different enantiomers of ketamine. Consider instead: "Ketamine is phenylcyclidine derivative with two optical enantiomers. It is commercially available either as a racemic mixture, or as the (S) enantiomer alone, which features approximately twice the potency of the racemic mixture with potentially fewer psychomimetic side effects." (i found some case reports on this, including "Paul R, Schaaff N, Padberg F, Möller HJ, Frodl T. Comparison of racemic ketamine and S-ketamine in treatment-resistant major depression: report of two cases. World J Biol Psychiatry. 2009;10(3):241-4. doi: 10.1080/15622970701714370. PMID: 19224412."). While you are describing the chemical characteristics of the drug, before getting into its use, this would be an ideal place to merge in the mechanism of action/ NMDA receptor antagonism, which was previously paragraph 3. From there you can more smoothly transition into the effects and benefits of the drug:

"Featuring sedative, amnestic, and analgesic properties, Ketamine has the potential to be a sole agent for induction and maintenance of anaesthesia."

Paragraph 4:

Much of the material at the beginning of this paragraph would serve you better in paragraph 1 or 2, where you already started to describe ketamine's multiple uses and favorable traits. I would consolidate. After the line ending in "maintenance of anaesthesia." in paragraph 1, you can expand a little with its multiple roles in current medical practice.. Instead of "the clinical use of ketamine is primarily for..." consider "Due to its sympathomimetic and bronchodilating effects, Ketamine has found favor internationally as a sedative hypnotic agent in patients with hemodynamic instability and those with reactive airway disease." You can add in the use in developing countries there as well as you close out the second paragraph. I would make a break/new paragraph that begins with your line "A number of national guidelines..." because this is where you bring the reader back to your purpose, an exploration of the merits of this multifaceted sedative and analgesic medicine in the context of brain injury patients.

Suggestions for "Why it is important"

This is where your article does a very nice job of drawing the reader in. I think you could enhance up the argument by starting this paragraph slightly stronger. Instead of "Ketamine has been used scarcely..." consider:

"Despite its multiple potential benefits, concerns of increased ICP has limited ketamine's use in patients with severe acute brain injury. Yet recent analyses suggest that ICP-related reservations about ketamine may be unfounded."

The rest of the paper read very well for me.

Overall this strikes me as a very thoughtful proposal and I'm eager to read your findings when the meta analysis is complete.

Reviewer #2: The authors propose a protocol for a systematic review focusing on use of ketamine in acute neurological injury. For all the many reasons the authors discuss, this is becoming a topic of high interest.

-The authors acknowledge that the literature on this topic will be highly heterogeneous in terms of patient population, study design, and outcomes assessed. They also acknowledge that they may analyze the results differently depending on the findings. Combining these heterogenous studies, which all used ketamine in a different way with different doses in different patient populations focusing on different outcomes seems like attempting to combine these results into a meaningful meta-analysis is of questionable validity. Knowing this literature moderately well at least, I do not think the authors will find any well done RCTs that would fit into the analysis approach they suggest. With strict rigorous criteria, do the authors think that there are additional published studies that will add to the systematic review from 2011 (Roberts) which identified no comparative studies?

-Given that there may need to be multiple different approaches depending on what the authors find and the bottom line is that this SR will most likely be focused on low quality data, what is the justification for publishing a protocol?

Reviewer #3: In this manuscript, dr. Andreasen and colleagues present the study protocol for a systematic review and meta-analysis of RCTs investigating the effect of ketamine in patients with traumatic brain injury.

The protocol deals with a very interesting topic, as in my experience use of ketamine for "neuro" patients is still considered contraindicated by many colleagues despite potential advantages.

The protocol is well written, and the study well planned. I only have few minor comments:

- I assume that PRISMA-P will be followed for this protocol and PRISMA 2020 for the full report (see p. 8, first Methods paragraph)

- I suggest to plan a subgroup/sensitivity analysis excluding crossover studies

7. PLOS authors have the option to publish the peer review history of their article (what does this mean?). If published, this will include your full peer review and any attached files.

Reviewer #1: **Yes: **Christopher Conley, MD

Reviewer #2: No

Reviewer #3: No

---

## [Author Response · Author response to Decision Letter 0]

4 Oct 2021

Title of the Manuscript: Ketamine for critically ill patients with severe acute brain injury: protocol for a systematic review with meta-analysis and Trial Sequential Analysis of randomised

clinical trials

Manuscript number: PONE-D-21-17647

Response to reviewers and Editor

We thank you for the opportunity to reply to the reviewers’ and Editor’s comments regarding our submission entitled “Ketamine for critically ill patients with severe acute brain injury: protocol for a systematic review with meta-analysis and Trial Sequential Analysis of randomised clinical trials” (PONE-D-21-17647). 

In this response, we have written the reviewer comment, followed by our response, and the text parts from the revised manuscript. The page references are to the Manuscript file without track changes. 

Responses to the Editor’s comments

Response: Changes have been made to the title page so that the formatting guidelines for title, authors, and affiliations are met. Likewise, changes in font size of headings have been made throughout the manuscript, as well as a few minor corrections to the wording, and the section Supporting information has been moved to after the reference list. 

Response: We have updated our information on funding. The authors received financial support from the Department of Neurosurgery Research Pool (Neurokirurgisk forskningspulje) at Rigshospitalet, Master Carpenter Sophus Jacobsen and wife Astrid Jacobsen’s Foundation (Snedkermester Sophus Jacobsen og hustru Astrid Jacobsen’s Fond), Rigshospitalet’s 3-year PhD scholarship, and the Danish Victims Foundation (20-610-00103) in the form of salary for author TA. Funding was also received from: The A.P. Møller Foundation (20-L-0041), Knud og Edith Eriksen’s Memorial Fund (Knud og Edith Eriksens Mindefond, 62786-2021), and The Novo Nordisk Foundation (NNF20OC0065750). The funders had no role in study design, data collection and analysis, decision to publish, or preparation of the manuscript. 

This information was also added to the cover letter.

Response: The ethics statement has been moved to the Methods section on page 8. 

Response: The reference list has been reviewed and no changes have been made. We assert that it stands complete and correct. 

5. Thank you for submitting the above manuscript to PLOS ONE. During our internal evaluation of the manuscript, we found significant text overlap between your submission and the following previously published works, some of which you are an author.

https://systematicreviewsjournal.biomedcentral.com/articles/10.1186/s13643-019-0957-0

https://link.springer.com/chapter/10.1007%2F978-981-13-7272-8_1

Please revise the manuscript to rephrase the duplicated text, cite your sources, and provide details as to how the current manuscript advances on previous work. Please note that further consideration is dependent on the submission of a manuscript that addresses these concerns about the overlap in text with published work.

Response: Thank you for making us aware of this text overlap. We have not intentionally copied text from other publications and agree that duplicating text is not acceptable. We have now read the two listed publications carefully to identify the text overlap. 

For the first publication, we found text overlap for Primary outcomes (page 9-10), in the Selection of studies and Data extraction and management sections (page 11-129, as well as the section Trial Sequential Analysis (page 16), and how continuous outcomes is managed under Sensitivity analysis (page 17). As mentioned, one of the authors of our protocol (CG) was also the co-author of the protocol entitled “Ivabradine for coronary artery disease and/or heart failure—a protocol for a systematic review of randomised clinical trials with meta-analysis and Trial Sequential Analysis”. The resemblance in text must therefore be considered text recycling. We as authors of the present protocol find that the resemblance in text is limited to certain phrases of the Methods section. Accuracy and precision in the description of the methods applied is key for comprehension and reproducibility. Therefore, text recycling is not uncommon in the mentioned sections, especially not for the same author, who may have evolved a precise wording of methodological definitions and practice. 

As an example, we share two of the same primary outcomes (quality of life and proportion of participants with one or more serious adverse events); both these outcomes are highly relevant, especially when dealing with patients suffering from acute severe brain injury. In our description of a serious adverse event, we have included the definition from the ICH (International Conference on Harmonization) since international consensus has been achieved around these criteria. This gives rise to overlap in text because the authors of the Ivabradine protocol also used this definition. 

In the description of the parameters we will use in the Trial Sequential Analysis (TSA), there is also some overlap in text. Nevertheless, this paragraph is a short and precise account of the methods chosen for the TSA. 

For the second publication, the chapter called “Sedation and Analgesia for Patients with Acute Brain Injury” from the book “Neurocritical Care”, we the authors of the present protocol were unaware of this publication before you directed our attention to it. Also, we cannot find any overlap in text. Please advise on any misunderstanding on our part. 

Minor comments from Editor:

1. I suggest including a PRISMA 2020 checklist and a PRISMA 2020 flow-chart in your systematic review.

Response: We have now clarified in the manuscript that a PRISMA 2020 checklist and a PRISMA 2020 flow-chart will be used in our systematic review.

"We have added the following on page 8: This protocol will adhere to the Preferred Reporting Items for Systematic Review and Meta-Analysis (PRISMA-P) checklist [71]. Likewise, the upcoming systematic review will be reported according to the PRISMA 2020 guidelines [72]. Inclusion and exclusion of trials will be visualised in a 2020 PRISMA flow diagram [72]." 

2. Please report more details on missing outcome data. What does “important missing data” mean? You will contact corresponding authors only for primary outcome data? What about, for example, secondary outcomes or methodological information for risk of bias? Probably you should define a priori some important missing data to be asked to corresponding authors. How many times will you contact corresponding authors in case of no reply?

Response: We have extended the “Missing data” section with the following paragraph, page 14:

"We will contact investigators to retrieve missing data. This applies to missing data for our primary, secondary, and exploratory outcomes, as well as methodological information for the risk of bias assessment. Additionally, we will seek information on the relevant subgroup in case of trials with a mixed population. In case of no reply, we will make a second and third attempt to get in contact with the authors in question. If the missing data is irretrievable, we will take the percentage of missing data into account when interpreting the results (see below). Standard deviations (SD), if not reported by authors, will be calculated using data from the trial if possible."

Responces to the reviewers’ comments

Reviewer #1

This proposed study protocol has sound methodology. This study proposal is certainly a question worth asking, with important clinical implications that could change the way we manage this vulnerable subset of patients. Some small changes to the language used may make it easier to follow. I've included some examples below:

Responses: We thank the reviewer for the kind words. We would also like to thank the reviewer for their time and work on this protocol.

Suggestions for the Introduction:

Have faith in your audience. Most would agree that severe acute brain injury generally has two initial pathways, admission to ICU or death. It doesn't add much to say they may be admitted to ICU. Instead your intro may be more effective if the first line read: "Intensive care for patients with severe acute brain injury often includes administration of sedative and analgesic medications."

Responses: We thank the reviewer for the suggestion. We have changed the text in the Introduction on page 3 as suggested.

The sentence beginning with "Existing guidelines" probably shouldn't read "do not recommend," as this may also be interpreted as "recommend against," which is not your intent. Instead you may consider: "Existing guidelines do not specify sedative and analgesic agents of choice in patients with severe acute brain injury. Propofol and midazolam are currently the most frequently used drugs in this setting."

Response: We have changed the text in the Introduction on page 3 accordingly.

Suggestions for "Description of the Patient Population"

Consider consolidating the first 2 sentences in this section: "Encompassing traumatic brain injury, subarachnoid haemorrhage, intracerebral haemorrhage, acute ischaemic stroke, hypoxic brain injury after cardiac arrest, as well as complications from neurosurgery, Severe acute brain injury is a common cause of critical illness requiring admission to intensive or neurointensive care units."

Response: We thank the reviewer. We have changed the text in the Introduction under “Description of the patient population” on page 3.

Paragraph 2: substitute "prolonged" for "long durations of"

Response: We have changed the text in the Introduction under “Description of the patient population” on page 3 as suggested.

Paragraph 3: You already described the range of specific diseases that severe acute brain injury encompasses. Consider removing the intro sentence beginning "the specific diseases." and delete the words "as examples." Instead begin the paragraph with "Aneurysmal subarachnoid hemorrhage, which occurs...."

Response: We have changed the text in the Introduction under “Description of the patient population” on page 3 as suggested.

After "long-term outcome", "Early aneurysm closure..." should be it's own new sentence (not a semicolon)

Response: We have changed the text in the Introduction under “Description of the patient population” on page 3 as suggested.

After "60%" consider ending the sentence and starting a new sentence. Consider "Only 12% to 39% of afflicted patients will achieve functional independence."

Response: We agree and have changed the text in the Introduction under “Description of the patient population” on page 4 as suggested.

Paragraph 4 (Treatment):

2nd sentence reads awkwardly. Consider instead, "Administration of nimodipine, for example, may reduce the risk of delayed cerebral ischemia..."

Response: We have changed the text in the Introduction under “Description of the patient population” on page 4 as suggested.

Suggestions for "Description of the intervention"

Paragraph 1: I would spend a little more time expanding here. (some physicians may also be less familiar with S-ketamine so there is an opportunity to educate. As written, it starts out a bit unclear when describing the different enantiomers of ketamine. Consider instead: "Ketamine is phenylcyclidine derivative with two optical enantiomers. It is commercially available either as a racemic mixture, or as the (S) enantiomer alone, which features approximately twice the potency of the racemic mixture with potentially fewer psychomimetic side effects." (i found some case reports on this, including "Paul R, Schaaff N, Padberg F, Möller HJ, Frodl T. Comparison of racemic ketamine and S-ketamine in treatment-resistant major depression: report of two cases. World J Biol Psychiatry. 2009;10(3):241-4. doi: 10.1080/15622970701714370. PMID: 19224412."). While you are describing the chemical characteristics of the drug, before getting into its use, this would be an ideal place to merge in the mechanism of action/ NMDA receptor antagonism, which was previously paragraph 3. From there you can more smoothly transition into the effects and benefits of the drug: "Featuring sedative, amnestic, and analgesic properties, Ketamine has the potential to be a sole agent for induction and maintenance of anaesthesia."

Response: We thank the reviewer for their thoughtful comments and suggestions. We have changed the text accordingly on page 5: 

"Ketamine is a phenylcyclidine derivate with two optical enantiomers [40]. It is commercially available either as a racemic mixture, or as the (S)-enantiomer alone, which features approximately twice the potency of the racemic mixture [41], with potentially fewer psychomimetic adverse effects [42]."

Reference 42 is the reference Paul et al. 2009 as proposed by the reviewer.

The text is followed by these two paragraphs, which were moved from page 5: 

"The main mechanism of action of ketamine has been attributed to NMDA-receptor (N-methyl-D-aspartate-receptor) antagonism [43]. The NMDA receptor is an ionotropic glutamate receptor. Activation of the receptor requires two distinct stimuli with temporal proximity – glutamate binding and membrane depolarisation [44]. 

Even though ketamine has a direct negative inotropic and chronotropic effect on the heart, ketamine has an overall sympathomimetic effect resulting in increased heart rate, cardiac output, and mean arterial blood pressure [45]. It is often stated that the cardiovascular stability of ketamine makes it a favourable choice in haemodynamically compromised patients, although there have been voices of dissent [46]." 

Followed by the suggested change on page 5:

"Featuring sedative, amnestic, and analgesic properties, ketamine has the potential to be the sole agent for induction and maintenance of anaesthesia [40]." 

Paragraph 4:

Much of the material at the beginning of this paragraph would serve you better in paragraph 1 or 2, where you already started to describe ketamine's multiple uses and favorable traits. I would consolidate. After the line ending in "maintenance of anaesthesia." in paragraph 1, you can expand a little with its multiple roles in current medical practice. Instead of "the clinical use of ketamine is primarily for..." consider "Due to its sympathomimetic and bronchodilating effects, Ketamine has found favor internationally as a sedative hypnotic agent in patients with hemodynamic instability and those with reactive airway disease." You can add in the use in developing countries there as well as you close out the second paragraph. I would make a break/new paragraph that begins with your line "A number of national guidelines..." because this is where you bring the reader back to your purpose, an exploration of the merits of this multifaceted sedative and analgesic medicine in the context of brain injury patients.

Response: We agree with the proposed changes and have rephrased the text as suggested by adding the following on page 5:

"Due to its sympathomimetic and bronchodilating effects, ketamine has found favour internationally as a sedative hypnotic agent in patients with hemodynamic instability and in those with reactive airway disease." 

We have also deleted the suggested sentence “The clinical use of ketamine is primarily for patients with haemodynamic instability and reactive airway disease.” and made a new paragraph before “A number of national guidelines...” on page 5-6.

Suggestions for "Why it is important"

This is where your article does a very nice job of drawing the reader in. I think you could enhance up the argument by starting this paragraph slightly stronger. Instead of "Ketamine has been used scarcely..." consider:

"Despite its multiple potential benefits, concerns of increased ICP has limited ketamine's use in patients with severe acute brain injury. Yet recent analyses suggest that ICP-related reservations about ketamine may be unfounded."

Response: We have changed the text accordingly by adding on page 7:

"Despite its multiple potential benefits, concerns of increased ICP has limited the use of ketamine in patients with severe acute brain injury [8,60]. Yet recent analyses suggest that ICP-related reservations about ketamine may be unfounded [61,62]."

We have deleted “Ketamine has been used scarcely in the ICU for patients with severe acute brain injury [8], as it was reported to increase ICP [59]. However, this was not confirmed by recent analyses.”

The rest of the paper read very well for me.

Overall this strikes me as a very thoughtful proposal and I'm eager to read your findings when the meta-analysis is complete.

Response: We thank the reviewer for their thoughtful and valuable comments.

Reviewer #2

The authors propose a protocol for a systematic review focusing on use of ketamine in acute neurological injury. For all the many reasons the authors discuss, this is becoming a topic of high interest.

- The authors acknowledge that the literature on this topic will be highly heterogeneous in terms of patient population, study design, and outcomes assessed. They also acknowledge that they may analyse the results differently depending on the findings. Combining these heterogenous studies, which all used ketamine in a different way with different doses in different patient populations focusing on different outcomes seems like attempting to combine these results into a meaningful meta-analysis is of questionable validity. Knowing this literature moderately well at least, I do not think the authors will find any well done RCTs that would fit into the analysis approach they suggest. With strict rigorous criteria, do the authors think that there are additional published studies that will add to the systematic review from 2011 (Roberts) which identified no comparative studies?

Response: We thank the reviewer for the thorough work and thoughtful comments to our protocol. We agree that the literature on this topic is highly heterogeneous. We believe that it is meaningful to explore the use of ketamine in patients with severe acute brain injury since these patients are in high risk of a poor outcome, and there is a need to improve the current standard of care. Ketamine is scarcely used in neurointensive care, although recent findings suggest that ketamine inhibits cortical spreading depolarisations, which may be causally related to lesion expansion in patients with acute severe brain injury. The review by Roberts et al. from 2011 on sedation for critically ill adults focuses mainly on traumatic brain injury patients. The present protocol focuses on trials with patients suffering from the full range of diseases known to be associated with cotical spreading depolarisation. This includes subarachnoid haemorrhage, traumatic brain injury, acute ischaemic stroke with massive hemispheric infarction, and spontaneous intracerebral haemorrhage with postoperative perihaematomal oedema progression. Since ketamine appears to inhibit these pathological electrophysical changes, it is exactly this patient population that may be worth examining. Additionally, the review by Roberts et al. was published 10 years ago, and new trials and studies trials could have emerged in the meantime, especially given the emerging interest in cortical spreading depolarisations. We think that it is important to approach this question in a systematic manner. This entails undertaking a systematic review. By nature, this includes conducting a systematic search and subsequently analysing and interpreting the pooled results through sound and robust methodology. That is why we have proposed this systematic review following the recommendations from the Cochrane Handbook for Systematic Reviews of Interventions and the eight-step assessment suggested by Jakobsen et al. Only by doing this, we can obtain definite evidence, or lack thereof, on the effect of ketamine in this vulnerable subset of patients. 

- Given that there may need to be multiple different approaches depending on what the authors find and the bottom line is that this SR will most likely be focused on low quality data, what is the justification for publishing a protocol?

Response: We chose to be transparent in our methodology and therefore seek to publish the protocol. By publishing the protocol, and making our planned methodology public, we intend to decrease the risk of bias for the systematic review and meta-analysis as we have thus committed ourselves to the specified methods before performing the review. 

Reviewer #3

In this manuscript, dr. Andreasen and colleagues present the study protocol for a systematic review and meta-analysis of RCTs investigating the effect of ketamine in patients with traumatic brain injury. The protocol deals with a very interesting topic, as in my experience use of ketamine for "neuro" patients is still considered contraindicated by many colleagues despite potential advantages. The protocol is well written, and the study well planned. I only have few minor comments:

- I assume that PRISMA-P will be followed for this protocol and PRISMA 2020 for the full report (see p. 8, first Methods paragraph)

Response: We thank the reviewer for the kind words as well as the time and work spent on this protocol. PRISMA-P will be followed for this protocol and PRISMA 2020 for the systematic review. It is now clarified in the revised manuscript on page 8:

"This protocol will adhere to the Preferred Reporting Items for Systematic Review and Meta-Analysis (PRISMA-P) checklist [71]. Likewise, the upcoming systematic review will be reported according to the PRISMA 2020 guidelines [72]. Inclusion and exclusion of trials will be visualised in a 2020 PRISMA flow diagram [72]." 

- I suggest planning a subgroup/sensitivity analysis excluding crossover studies

Response: We thank the reviewer for this suggestion, but as we only intend to use data of the first period of the cross over trials (during which they resemble all other randomised trials) we are not sure that this sensitivity may be meaningful. We have now clarified that we only use data from the first period of such trials on page 13:

"For cross-over trials, we will include only outcomes from the end of the first period of the trial where cross-over trials resemble usual randomised clinical trials, to avoid bias arising from carry-over effects."

We would like to thank both the reviewers and the Editor for consideration and thorough reading of our manuscript, and for the constructive comments and suggestions, which have helped us improve the manuscript. This is hereby resubmitted to PLOS ONE.

On behalf of all authors,

Yours sincerely 

Trine Hjorslev Andreasen, MD

---

## [Decision Letter · Decision Letter 1]

29 Oct 2021

Ketamine for critically ill patients with severe acute brain injury: protocol for a systematic review with meta-analysis and Trial Sequential Analysis of randomised clinical trials

PONE-D-21-17647R1

Dear Dr. Andreasen,

Thank you for your further work on this manuscript which now makes a fine contribution to consideration of this important topic.

We’re pleased to inform you that your manuscript has been judged scientifically suitable for publication and will be formally accepted for publication once it meets all outstanding technical requirements.

Kind regards,

Alessandro Putzu, M.D.

Academic Editor

PLOS ONE

Reviewers' comments:

Reviewer's Responses to Questions

**Comments to the Author**

1. Does the manuscript provide a valid rationale for the proposed study, with clearly identified and justified research questions?

Reviewer #1: Yes

Reviewer #3: Yes

2. Is the protocol technically sound and planned in a manner that will lead to a meaningful outcome and allow testing the stated hypotheses?

Reviewer #1: Yes

Reviewer #3: Yes

3. Is the methodology feasible and described in sufficient detail to allow the work to be replicable?

Reviewer #1: Yes

Reviewer #3: Yes

4. Have the authors described where all data underlying the findings will be made available when the study is complete?

Reviewer #1: Yes

Reviewer #3: Yes

5. Is the manuscript presented in an intelligible fashion and written in standard English?

Reviewer #1: Yes

Reviewer #3: Yes

6. Review Comments to the Author

You may also provide optional suggestions and comments to authors that they might find helpful in planning their study.

Reviewer #1: The manuscript reads better now. Thank you for incorporating my feedback. Hopefully this article will catalyze further study of ketamine's potential in this vulnerable patient population.

Reviewer #3: In this manuscript, dr. Andreasen and colleagues now present a revised version of their work.

I believe that all of my comments have been adequately addressed and the manuscript is in my opinion now suitable for publication.

7. PLOS authors have the option to publish the peer review history of their article (what does this mean?). If published, this will include your full peer review and any attached files.

Reviewer #1: No

Reviewer #3: No

---

## [Editor Report · Acceptance letter]

4 Nov 2021

PONE-D-21-17647R1 

Ketamine for critically ill patients with severe acute brain injury: protocol for a systematic review with meta-analysis and Trial Sequential Analysis of randomised clinical trials 

Dear Dr. Andreasen:

I'm pleased to inform you that your manuscript has been deemed suitable for publication in PLOS ONE. Congratulations! Your manuscript is now with our production department. 

Kind regards, 

on behalf of

Dr. Alessandro Putzu 

Academic Editor

PLOS ONE